

# Rainfall enhancement downwind of hills due to standing waves on the melting-level and the extreme rainfall of December 2015 in the Lake District of northwest England

Edward Carroll[1]

[1]Independent researcher, Garden Cottage, Farringdon, Exeter EX5 2JA, UK

*Correspondence to*: Edward Carroll (edward_carroll@hotmail.co.uk)

**Abstract.** Flow over orography can be investigated through stationary gravity waves, i.e. those whose speed exactly opposes, and therefore cancels, that of the airstream in which they are embedded. They give rise to persistent zones of ascent and

descent, which modulate precipitation patterns and contribute to large accumulations, e.g. through the well-known seeder-feeder mechanism. It is shown here that opposite, stationary waves on the melting-level focus rain, potentially multiplying intensity downwind of hills by a factor of rain fall speed divided by snow fall speed, and that the effect is maximised when the vertical profile near the melting-level is isothermal. A 2D diagnostic model based on linear gravity wave theory is used to investigate the record-breaking rainfall of December 2015 in the Lake District of northwest England. The pattern of vertical

velocity is shown to have a good, qualitative fit to that of the Met Office's operational, high-resolution UKV model averaged over 24 hours, suggesting that orographically excited standing waves were the principal cause of the rain. Precipitation trajectories suggest that a persistent, downstream, elevated wave caused by the Isle of Man maintained a spray of seeding ice particles directed towards the Lake District; that these grew whilst suspended in strong upslope flow before being focussed by the undulating melting-level into intense shafts of rain.

**1 Introduction**

It has long been observed that orography has a profound influence on rainfall enhancement and distribution, with ranges of hills, especially those exposed to the prevailing wind, often being wetter than lower-lying or more sheltered areas. Given the importance of spatial distribution of rainfall e.g. to river flow modelling and flood warning systems, mechanisms of enhancement have been much studied conceptually, observationally, analytically and with numerical models, for the UK and

more widely. One of the first to be described is the seeder-feeder effect (Bergeron, 1965), whereby pre-existing, seeding raindrops sweep out cloud droplets formed in a capping, or feeder, cloud which is continuously replenished by the ascent of air over a hill. Since then many authors have dealt with the mechanism, Lean and Browning (2013) providing a useful review.

Explanations of the seeder-feeder effect generally invoke some non-orographic origin to the independently existing seeder

precipitation, such as frontal ascent, though Browning et al. (1974) discuss self-seeding by hills via release of potential



instability; whilst Robichaud and Austin (1988) point out that hills of long wavelength (half width more than about 20 km, depending on wind speed) will allow a large enough residence time for cloud droplets to grow to precipitation size within the capping cloud itself, leading to what is effectively a separate category of orographic enhancement. Whilst the precipitation maximum in orographically enhanced rain tends to occur some way up the slope on the windward side of the hill, the effects

of wind drift can cause the largest amounts to fall beyond the summit of a hill or range of hills, as found e.g. by Hobbs et al., (1973), Lean and Browning (2013).

Other authors, such as Bruintjes et al. (1994), have pointed to the importance of stationary gravity waves in determining the distribution of precipitation in areas of complex terrain. Colle (2008) used two-dimensional simulations to show how the

upstream tilt of a gravity wave from a hill crest can displace precipitation distribution in this direction, whilst Stout et al. (1993) found that differential advection in gravity wave motion preferentially concentrates particle deposition in distinct zones.

In this paper it is shown graphically and analytically that there is an important mechanism, depending partly on differential wind drift of precipitation in gravity waves and, more significantly, on gravity wave induced undulations in the melting layer,

which can enhance rainfall significantly downwind of high ground. Investigations with a two-dimensional gravity wave model, along with output from an operational, high resolution, non-hydrostatic, primitive equation model suggest that elevated wave generated by an isolated hill well upstream could be influential in acting as a rich source of ice seeds. It is advanced that both these elements are likely to have been influential in the extreme rainfall recorded at Honister Pass, Cumbria, in December 2015.

**2 Theory**

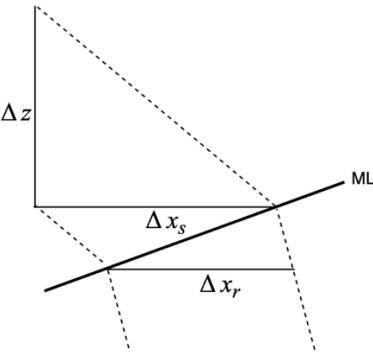

**Figure 1: A sloping melting-level (ML) dividing streams of falling snow from rain, with horizontal wind blowing from left.**






Consider snow with fall speed $w_s$ in an environment of uniform horizontal velocity $U$ which turns to rain on encountering the melting-level. If the latter is fixed and sloping upwards along the flow, the intensity of precipitation is increased since the resulting rain is spread over a smaller horizontal area, as can be seen in Fig. 1, whereas with a horizontal melting level, as in Fig. 2(a), the intensity of precipitation is the same above and below the melting level. The enhancement factor, $E$, can be given

by the ratio of the two cross-sectional areas over which precipitation generated in the column of height $\Delta z$ falls:

$$E = \frac{\Delta x_s}{\Delta x_r}$$

We use the identity

$$\frac{\Delta x_s}{\Delta x_r} = \frac{\Delta z}{\Delta x_r} \bigg/ \frac{\Delta z}{\Delta x_s}$$

and the fact that in a base situation of no vertical wind and horizontal melting level (as in Fig. 2(a)) snow particles fall with a slope given by

$$\frac{\Delta z}{\Delta x_s} = \frac{-w_s}{U}$$

Note that fall speeds are treated as negative because they are in the direction of decreasing $z$. Rearranging and taking limits, as $\Delta \to 0$, we can give the enhancement, $E$, of rain rate over the base situation of no vertical wind and horizontal melting-level as

$$E = -\frac{\partial z}{\partial x_r} \frac{U}{w_s}$$

$$(1)$$

Melting of snow is depicted in Fig. 1 and in subsequent diagrams as instantaneous in order to simplify the analysis, though the same result would be achieved by allowing a layer over which a gradual transition in fall speed occurred.





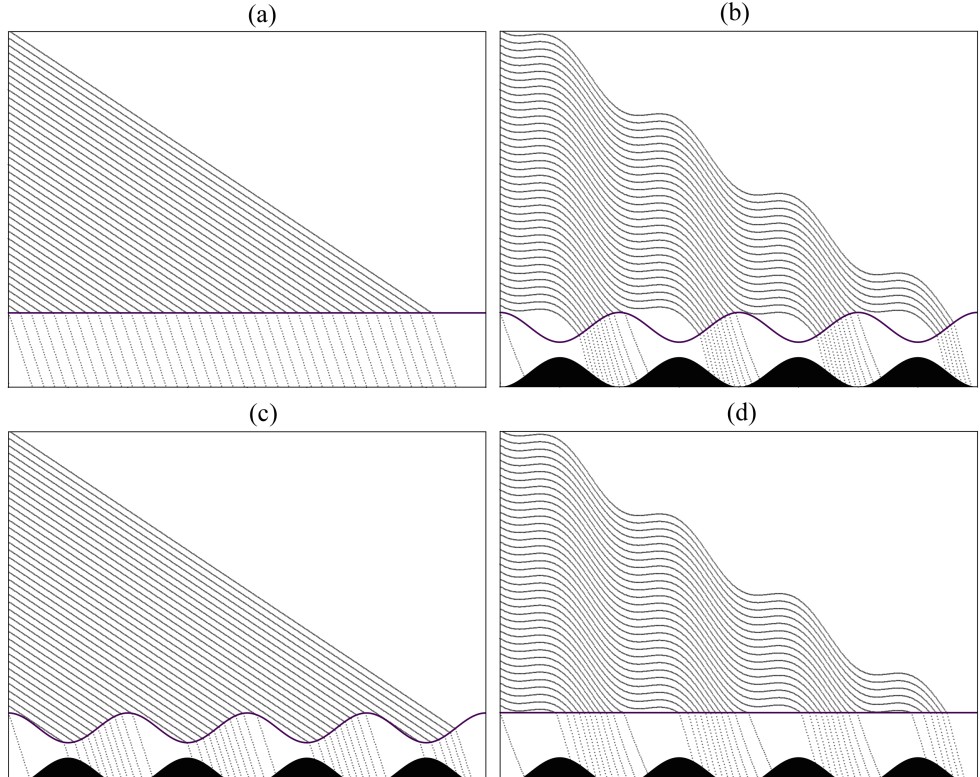

**Figure 2: Falling snow turning to rain with wind from the left and different configurations of melting-level and vertical velocity. See**
**text for explanations.**

In Fig. 2 are four different scenarios based on a wind blowing falling precipitation from the left. The melting-level remains invariant with time and topography, where it is not flat, is shown in black. Precipitation particle trajectories shown were calculated with a simple Lagrangian model.

In 2(a), which is considered as the base state, there is no vertical wind, a horizontal melting-level. Horizontal distances between streams of falling rain and snow are the same and the precipitation rate remains unenhanced and uniform at the base rate.

In 2(b), a range of hills gives rise to stationary gravity waves which modulate the melting-level, lowering it through adiabatic cooling in areas of ascent, raising it in areas of descent. (In this case the environmental lapse rate is half the adiabatic lapse
rate. It is shown later that this gives the melting-level variation the same amplitude as that of the wave motion). Precipitation is concentrated into discrete zones of heavier rain separating dry zones. The relatively low melting-level depicted here causes rainfall at the ground to be most concentrated on the lee side of the hill, though this is not an intrinsic characteristic of this scenario since a higher melting-level could cause it to fall on the windward side of downstream hills. Rainfall modulation can be seen as the aggregated effect of the two scenarios in (c) and (d).






In 2(c), there is no vertical wind but sinusoidally varying diabatic heating and cooling causes the melting-level to undulate, giving rise to dry and varyingly wet zones as snow is preferentially intercepted by the melting-level where it slopes upwards along the flow. This will be referred to as the melting-level slope effect.

In 2(d), there is gravity wave motion, but also sinusoidally varying, diabatic heating and cooling which exactly counteracts adiabatic temperature change, leaving a horizontal melting-level; a very unrealistic scenario, but one which is contrived to highlight a separate modulating mechanism whereby zones of ascent delay the descent of snow, causing it to become more concentrated in the descending phases. This will be referred to in this article as the bunching effect and is similar to the mechanism investigated by Stout et al. (1993), though theirs lacked a melting transition.


To examine these scenarios, we take the simplest case in which the flow pattern is constant with height, as would pertain when the orographic wavelength is equal to the critical wavelength (explained in section 5). As in studies such as those of Hobbs et al. (1973), it is assumed that the precipitation particle moves relative to the airstream within which it is embedded only downwards and at its terminal velocity. In doing so, any unbalanced aerodynamic forces are neglected. Use is made of a

stability parameter $\gamma$ defined as the environmental lapse rate expressed as a proportion of the adiabatic lapse rate, i.e.

$$\gamma = \frac{-\partial T/\partial z}{\Gamma},$$

where $\Gamma$ is the adiabatic lapse rate, saturated or dry according to the circumstance.


Since there is a discontinuity at the melting-level, the approach taken is to apply Eq. (1) in two stages – firstly to calculate enhancement along the melting-level, and extend it thence down to the ground. Derivations are given in Appendix A with the results summarised here:

The enhancement over base rate due to the combined bunching and melting-level slope is given by

$$E_g = \left(\frac{w_r + w_g}{w_s}\right) \left(\frac{w_m + \gamma w_s}{w_m + \gamma w_r}\right),$$

(2)





where $w_g$ is the vertical wind at the ground due to the slope of the terrain in the direction of the horizontal wind and $w_m$ is the vertical wind experienced by a precipitation particle at the melting-level. As before, $w_r$ and $w_s$ are the fall speeds of rain and snow respectively.

So the magnitude of modulation at the surface depends on the fall speeds of rain and snow as well as the phase and amplitude

of waves at the melting-level and ground. As the vertical temperature profile around the melting layer approaches isothermal, $\gamma \rightarrow 0$, the term in the vertical wind at the melting-level cancels and

$$E_g \rightarrow \frac{w_r + w_g}{w_s}$$

(3)


Since $w_g$, the vertical wind at the ground, is normally small compared with $w_r$, and persistent melting snow inclines the vertical temperature profile towards isothermal, we can state the first-order relationship as

$$E_g \approx \frac{w_r}{w_s}$$

(4)

Taking the example of snow fall speed of –1 m s$^{-1}$, a rain fall speed of –5 m s$^{-1}$, enhancement at the ground tends towards a factor of 5 given a near-isothermal environment around the melting-level. If there is +0.2 m s$^{-1}$ (ascent) at the ground due to motion up the slope of a downstream hill, this reduces enhancement to a factor of 4.8, whilst –0.2 m s$^{-1}$ gives 5.2. Of course,

these enhancement values are additional to those due to cloud physics processes such as feeder cloud washout.

The equations have been tested and found to agree with results from a simple, time-integrated, Lagrangian finite difference model which computes trajectories of falling and melting precipitation under different conditions of horizontal wind, stability and wave amplitude. It is also shown in Appendix A that the melting-level slope and bunching effects are separately equal

when $\gamma = 0.5$ (the situation depicted in Fig. 2), and that at higher static stability the melting-level slope effect dominates, equalling the full effect at $\gamma = 0$ (isothermal).

## 3 Discussion

The melting-level focussing mechanism has some similarities to the differential refraction of light by standing surface water waves which leads to patterns of light and dark on the bottom of a stream. This can be used as an analogy, whilst recognising

that fundamentally different physical mechanisms are at play.



The enhancement equations relate to situations in which the thermal pattern is fixed, with air flowing through the isotherms, as in stationary gravity waves. They are not applicable to structures such as frontal surfaces, where the thermal gradients move largely with the wind and are therefore crossed neither by air parcels, nor, horizontally, by precipitation particles. Indeed with the frontal case, at least in the frontogenetic phase, the transverse, thermally direct circulation (Eliassen, 1962) means that the slope of the melting-level bears an opposite relationship to the vertical velocity, with the higher melting-level, warmer air associated with ascent.

Melting snow extracts from the environment the heat necessary to effect the phase change from ice to water, resulting in a cooling tendency through the melting layer towards 0 C which can frequently lead to an isothermal or near-isothermal layer several hundred metres deep (Findeisen, 1940, Kain et al., 2000). Even modest warming from forced descent, e.g. down the lee side of a hill, in these circumstances will produce a steep downwind rise in the melting-level. The resulting melting-level slope effect will cause rapid interception of snow by the melting-level, leading to enhancement of rain at the surface which can be approximated by the rain fall speed divided by the snow fall speed. Even in situations where the cumulative effect of chilling in the melting layer is not enough to turn the profile isothermal, it will act to increase $\gamma$ and therefore enhancement according to Eq. (2).

The enhancement equations derived relate strictly only to an environment with constant horizontal wind speed and with flow patterns which do not vary in the vertical. In the context of wave motion, this is the simplest type of wave, whose phase remains constant in the vertical and which exhibits no change of amplitude with height, consistent with the critical wavelength (explained in section 5). Stout et al. (1993) did not deal with a melting transition but found in their treatment of the concentrating effect of the advection field that intensity was increased by upstream tilt of a wave with height and decreased by evanescence, i.e. reduction of wave amplitude with height. If only considering the enhancement across the melting-level, Eq. (2) is not invalidated by evanescence, since for the melting-level slope effect, which dominates at smaller $\gamma$, it is only amplitude at the melting-level which matters.

Because peak vertical velocity is proportional to wavenumber ($k$) through the relationship $w_{max} = \pm kAU$, where $A$ is wave amplitude and $U$ horizontal wind, the largest vertical velocities tend to be driven by the shorter waves. However, these waves, being generally smaller than the critical wavelength, lose amplitude with height through exponential evanescence. Therefore the lower the melting-level, the greater the enhancement potential through larger values of $w_m$ in Eq. (2). A lower melting-level also results in less rain-drift, keeping rain tied more to the lee slopes and so bringing a positive contribution to enhancement from $w_g$.



Since air cannot flow into the ground, $w_g$ is only non-zero where the ground slopes along the flow, this component contributing
to enhancement on lee slopes and reduction on windward slopes. Note that this treatment has dealt with rain which falls onto
ground whose contours follow the shape of the airflow. In the case of trapped lee waves, the pattern may persist well downwind
of the forcing topography, and with it the enhancement.

## 4 Gravity wave model (GWM)

A two-dimensional model was formulated for the purposes of examining the case of severe flooding in the Lake District, based
on diagnosis of standing waves, i.e. those whose upstream phase velocity is equal and opposite to the horiztonal wind ($U$) and
so maintain their position relative to the forcing orography. The driving equations come out of early work such as that of
Queney (1948) who, drawing together earlier studies, used the linearised momentum, continuity and thermodynamic equations
to find wave solutions which yield a phase velocity which depends on wavelength and stability. Singling out waves whose
phase velocity is equal and opposite to the flow within which they are embedded leads to following equations for stationary
waves:

$$w = W \exp\left(i(kx + mz)\right)$$

(5)

$$m^2 = N^2/U^2 - k^2$$


(6)

Where $k$ and $m$ are horizontal and vertical wavenumbers and $N$ is the buoyancy frequency given by

$$N = \sqrt{\frac{g}{T}\left(\frac{\partial T}{\partial z} + \Gamma\right)}$$


(7)

In the model, orography is decomposed into sinusoidal Fourier components, each exciting standing waves. For each, the
boundary value $W$, the vertical velocity forced at the surface, is derived from the product of the horizontal wind and the slope
of the terrain component. The stationary wave's vertical wavenumber $m$ is calculated from Eq. (6) and fed into Eq. (5), along
with the $k$ value appropriate to the Fourier component, to extend $W$ upwards from the surface. Amplitude is increased by a
factor of $\sqrt{\rho_s/\rho}$ where $\rho$ is density and $\rho_s$ surface density. Positive $m^2$ leads to proportionality with cos $mz$ through Euler's
formula, and a wave tilt from the vertical given by $m/k$; but if $m^2$ is negative, it results in a term in exp ($-mz$) and vertical
decay (evanescence). Contributions from each Fourier component are combined to arrive at a field of vertical velocity in the



*x-z* plane. The group velocity effects fall out via constructive interference of the resulting waves. The model allows the option of reinstating two terms which have been lost to approximations on the way to Eq. (6). One is in *f* and represents the horizontal restoring force arising from earth rotation. Its inclusion makes the waves intertia-gravity rather than purely gravity. Another takes account of the curvature in the vertical of the horizontal wind *U*, as in the Scorer parameter (Scorer, 1949).

The model was run with 20 vertical levels at a spacing of 500 m on terrain-following coordinates, and at a maximum wavenumber of 60 along its 300 km length, giving a smallest resolved wavelength of 5 km. Winds, temperatures and buoyancy frequency $N^2$, as well as being averaged over 24 hours, are also spatially averaged on vertical levels for the entire 300 km cross section, so vary only with height. Only the wind component in the plane of the cross section is used. There is a difficulty in specifying a suitable value of *N*, static stability being a key parameter in determining the structure of waves. The standard

expression assumes an unsaturated atmosphere, in which case the dry adiabatic lapse rate is used for Γ in Eq. (7). However, with such large values of rainfall in the case study, it was considered that latent heat release should not be ignored. Whilst it is possible to specify the equivalent buoyancy frequency for a saturated atmosphere, unsaturated layers will occur, especially in zones of descent, and even in largely saturated regions, inhomogeneity of humidity at smaller spatial and temporal scales will likely result in some intermediate value being appropriate to a treatment of time-averaged fields. A pragmatic choice was made

to use a value of *N* in which Γ in Eq. (7) takes a value part way between the saturated and unsaturated value, experiments showing that a 55% - 45% weighted mean of moist and dry values produced a good result. Note that for full accuracy, virtual temperatures should be used in the calculation of both dry and moist Γ, and that terms in the vertical variation in total water mixing ratio should be included in moist Γ (see Durran and Klemp, 1982). However, it is considered that at the temperatures in the case study these terms would be very small.


There are many theoretical limitations associated with the application of this model to the case under investigation. Perhaps least of these is the linearised nature of the governing equations, which are theoretically most valid for the lowest hills, though still form a reasonable approximation if Froude number is greater than one (Smith, 1980). In the case study presented, Froude number for the lower layers, given by *U* /*NH*, where *H* is hill height is around three. Non-linearities associated the inherent

unsteadiness of trapped of waves (see e.g. Durran, 2003) could also create some problems, and there is some sign of energy trapping downwind of the Isle of Man at about 6000 m. A more serious potential shortcoming is that the model's two dimensionality means that hills are treated as infinite ridges regardless of their extent normal to the cross section. A related issue is that the effect of orography lying either side of the line of the cross section, such as the lateral deflection of waves (Smith, 1980, Scorer, 1956), is not accounted for. Furthermore, wind components normal to the cross section may in reality

force a lower boundary vertical velocity at odds with that of the 2D model, especially where terrain varies most in this direction. In spite of the foregoing, the output of the model, when compared with that from the UKV, matched reasonably well as shown in the following section.



## 5 Case study

In the 24-hour period up to 18:00 UTC on 5 December 2015, 341.4 mm fell at Honister Pass (hereafter referred to as Honister)
in the Lake District of Cumbria, northwest England, setting a new UK 24-hour rainfall record, return period calculations
suggesting this to be more unusual than a one-in-a-thousand-year event (Marsh et al., 2016). This, and exceptionally high
rainfall totals recorded by other nearby gauges in the catchment of the river Derwent resulted in severe flooding in
Cockermouth and Keswick. 16 km further east, Brotherswater in the catchment of the river Eden recorded 293 mm in 24 hours,
this river registering its highest level on record, causing flooding in Carlisle (Marsh et al., 2016).


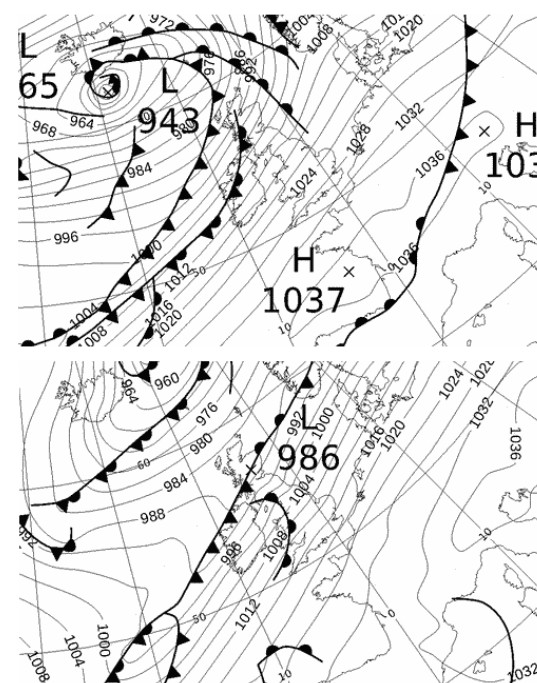

**Figure 3: Met Office analysis charts 18:00 UTC on 4 (top) and 5 (bottom) December 2015.**

A slow-moving pattern maintained a warm, moist airstream over the area through the period, originating at low latitudes in the
North Atlantic. Lying on the warm side of the polar front, thermal wind was relatively weak, and winds through much of the
troposphere were unidirectional and strong. An exceptionally strong zonal index was present at 18:00 UTC on 4 December,
evidenced by a low south of Iceland of 943 hPa and a high in the Bay of Biscay of 1037 hPa, the flow strength and direction
showing little change over England through the period. It may be inferred that the secondary, inner frontal zones analysed
crossing the area of interest are likely to have contributed deep layers of moisture. Synoptic-scale dynamical forcing for ascent
does not appear to have been high, but the long, narrow corridor of moisture extending from the Atlantic over the UK can be
identified with a warm conveyor belt / atmospheric river type structure (Dacre et al., 2019). Hourly rainfall totals at Honister





for the wettest 24 hour showed a remarkable uniformity suggesting a steady-state mechanism rather than the passage of transient forcings, and the prolonged nature of the event seems to be a key reason for its severity.

Data available for this study were in the form of 24-hour mean fields from the operational 09:00 UTC run of the Met Office's UKV model 4 December 2015. This is a 1.5 km grid length formulation of the non-hydrostatic, primitive equation Unified Model (Cullen et al., 1997, Sheridan et al., 2017). Output was extracted principally in the form of cross sections of mean values for the 24 hours to 18:00 UTC 5 December. These were aligned along the mean wind above the boundary layer through Honister and consisted of horizontal and vertical wind, along with temperature, at full horizontal resolution but only on 14 vertical levels. Data relating to moist variables were not available to the author.

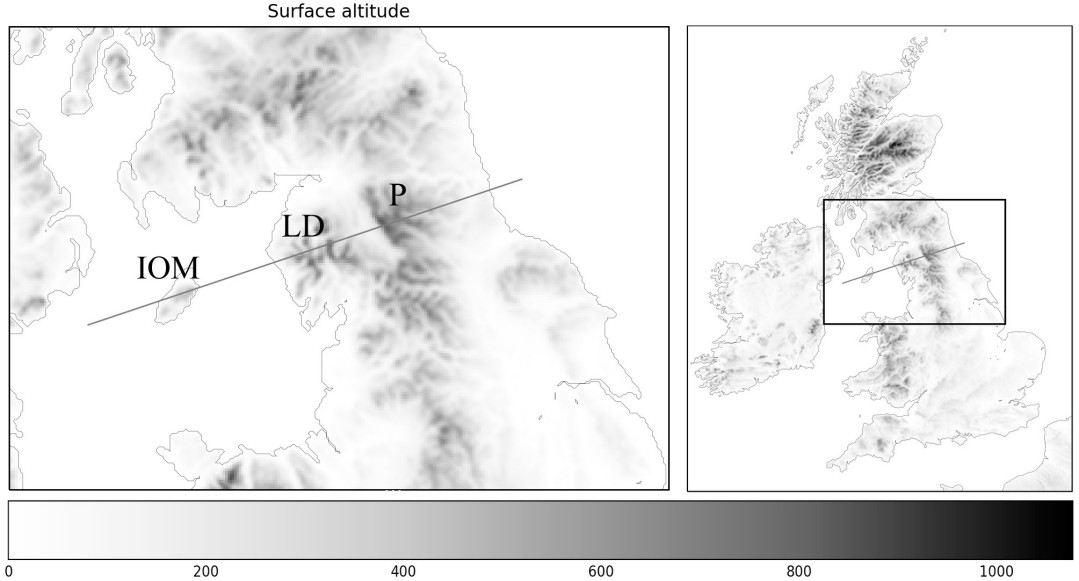

**Figure 4: Plan projection of cross section I on the UKV orography, a 200 km section of the full 300 km domain of the 2D gravity wave model. IOM, LD and P show the positions respectively of the Isle of Man, the Lake District and the Pennines.**





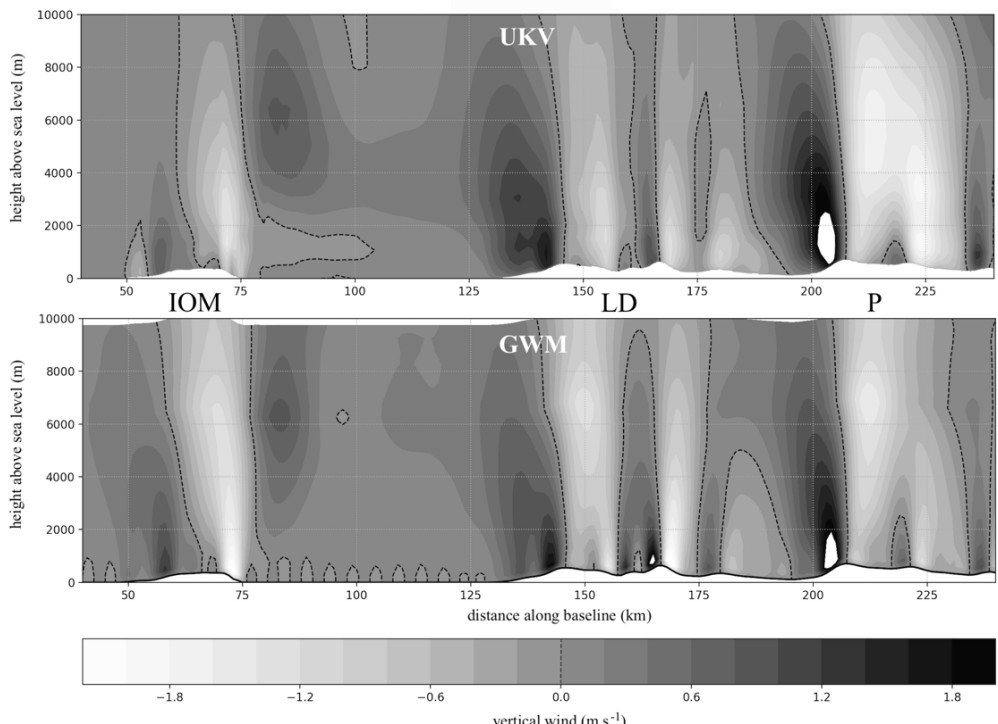

**Figure 5: Top: vertical velocity ($w$) in m s⁻¹ from UKV run from 09:00 UTC 4 December 2015 averaged over the 24 hours to 18:00**
**UTC 5 December 2015 along line of cross section shown in Fig. 4. Values above 2 m s⁻¹ are unfilled.**

**Bottom: As top but from 2D gravity wave model (GWM).**

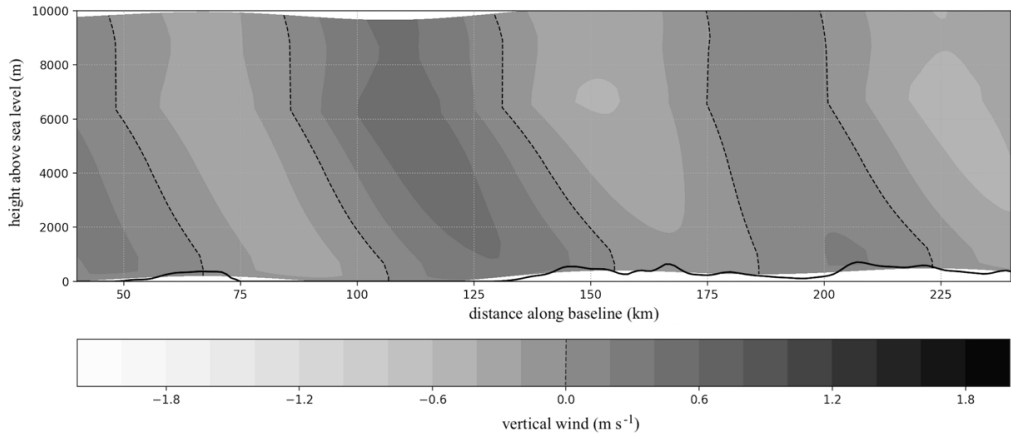

**Figure 6: Output from gravity wave model as in Fig. 5 but run at coarse resolution, with smallest resolved wavelength 77 km, the orographic wavelength of greatest amplitude in a Fourier decomposition. UKV orography is shown by black line. Gravity wave**
**model orography at this resolution in white.**



### 5.1 Discussion of gravity wave structure

Note that only the region from 40 to 240 km is shown in the cross section figures. The simple gravity wave model will be referred to as GWM, the Met Office operational 1.5 km model as UKV.

As a consequence of Eqs. (5) and (6), there exists a critical wavelength $L = 2\pi U / N$ below which waves are vertically aligned and decay in amplitude with height (i.e. exhibit evanescence). Stationary wave components of greater wavelength have an upstream tilt of wave phase with height and downwind transmission of energy along the group velocity vector which slopes upwards at $m/k$ from the horizontal. Values of $U$ and $N$ in the lower layers give L ~ 22 km. The Isle of Man, inducing a 30 km wavelength stationary wave, transmits wave energy downstream at a slope of $m/k$ ~ 0.8. However, a marked reduction in
stability above 6000 m (seen in Fig. 8) means this energy becomes trapped, leading to a maximum amplitude in the downwind portion at $x = 85$ km, $z = 6000$ m, where both UKV and GWM produce over 0.8 m s$^{-1}$.

In the context of frontal ascent, this value would be considered large, e.g. Heymsfield (1977) reported values of 0.01 – 0.1 m s$^{-1}$ in warm frontal settings, 0.2 m s$^{-1}$ in warm occluded fronts and 0.5 m s$^{-1}$ in convective regions of fronts. The vertical velocity
pattern is characteristic of the response to an isolated hill (e.g. Vosper 2015, Clark and Peltier, 1977), and can result in a stream of cold-topped cloud downwind of a hill, often with a sharp upwind edge. When seen in satellite imagery it is a familiar sign to operational meteorologists of a standing wave, though images from this case show any such cloud to be hidden by a shroud of frontal cloud. Elsewhere across the domain an elevated zone of ascent of this sort is not in evidence, presumably because of interference by adjacent features in the more complex orography of the Lake District and Pennines where the evanescent,
short wavelengths dominate.

Trapping leads to a low-amplitude oscillation in GWM extending further downwind of the Isle of Man at 6000 m manifested in a weak minimum at $x = 95$ and a weak maximum $x = 110$ which do not have counterparts in the UKV and could be a sign of the unreliability of the linearised formulation where such trapping is concerned.


The low-amplitude, surface-based undulation apparent either side of the Isle of Man in GWM is at the scale of the shortest resolved wavelength (5 km) and is considered to be grid scale noise, an artefact of the numerical treatment. Being of short wavelength, it is evanescent.

Fourier analysis of the orography for the whole of the domain shows a peak in amplitude at a wavelength of 77 km, representing the coarse components of the Isle of Man, Lake District and the Pennines. Being of long wavelength, it generates a rearward-tilting zone of ascent extending back over the sea towards the Isle of Man, shown more clearly in the coarse resolution GWM





experiment in Fig. 6. The abrupt change to vertically aligned phase seen in this figure is due to a transition to lower static stability aloft associated with the trapping of wave energy previously referred to.


Experiments were conducted with the GWM in which orographic features were selectively excluded in order to attribute the individual, at-a-distance influence of the Isle of Man, Lake District and Pennines. These confirm that the Lake District and, to a smaller degree the Pennines, contributed to a sloping zone of ascent tilting back across the Irish Sea towards the Isle of Man.

Running the GWM in full, inertia-gravity mode by including previously neglected terms in Coriolis parameter, $f$, and with the curvature term in the Scorer parameter reinstated, results in only very minor differences.

Some detail differences are seen between the two models, especially over the eastern side of the Lake District where the orography varies strongly perpendicular to the baseline of cross section amplifying the effect of the small normal wind 335 component. However, the generally good correspondence between this output and the UKV, despite all the caveats, along with the remarkable steadiness of rain rate at Honister, strongly suggests that gravity waves were the main driver for vertical velocity over the period of extreme rainfall and therefore for the rain itself. The zone of elevated ascent at $x = 85$, $z = 6000$ caused by the Isle of Man is thought to be important to rainfall over the Lake District and will be examined in the next section.

### 5.2 Precipitation trajectories and enhancement

Rather than attempting to model precipitation growth mechanisms, this study focuses on establishing plausible, representative precipitation trajectories to allow informed speculation on the provenance of the precipitation which fell in the Honister area, as well as enhancement over baseline rates due to bunching and melting-level slope effects previously elaborated.

The column of sustained ascent downwind of the Isle of Man might be expected to generate significant cloud and precipitation, 345 though to a degree difficult to quantify in the absence of moist variables from the UKV. Indeed, examining an extended cross section upstream this column seems the only region of ascent which could account for the rainfall in the Lake District; the strong ascent at lower levels immediately over the windward hills of the Lake District is too close to allow sufficient time for precipitation to form, though no doubt very important in the final stages of growth. It should be borne in mind that there is lee descent over the Isle of Man which is likely to have dried the air somewhat, though evaporation of pre-existing cloud water or 350 ice would mitigate this effect. In order to gain some measure of its precipitation-producing potential, a field was calculated of vertical velocity multiplied by the vertical gradient of volumetric saturated specific humidity along a saturated adiabat, to estimate the rate of condensation of water and/or deposition of ice in saturated unit volume (Fig. 7). Because of the reduction with temperature of water vapour carrying capacity, it gives a peak at lower levels than that of the vertical velocity field, centred around 4000 m and a temperature of – 6 C.




To address the potential limiting effect on condensation of upstream descent, which is stronger at lower levels, parcel trajectories were run from $x = 0$ to $x = 85$ with regularly spaced starting heights. The change in cloud water/ice content was calculated on the basis of integrated parcel height loss or gain multiplied by the vertical gradient of volumetric saturation specific humidity along a saturated adiabat (saturation here is with respect to water rather than ice). This suggests that with

saturated conditions at $x = 0$, parcels above 3250 m in the column at $x = 85$ gain cloud water and/or become more supersaturated despite the upstream descent. It is worth noting that in general, the greater the tilt of the wave, the less symmetrical it is about a vertical axis and the less the integrated ascent will be cancelled out by descent; this situation means that greater net condensation can be achieved by waves which exceed the critical wavelength more, other factors being equal.

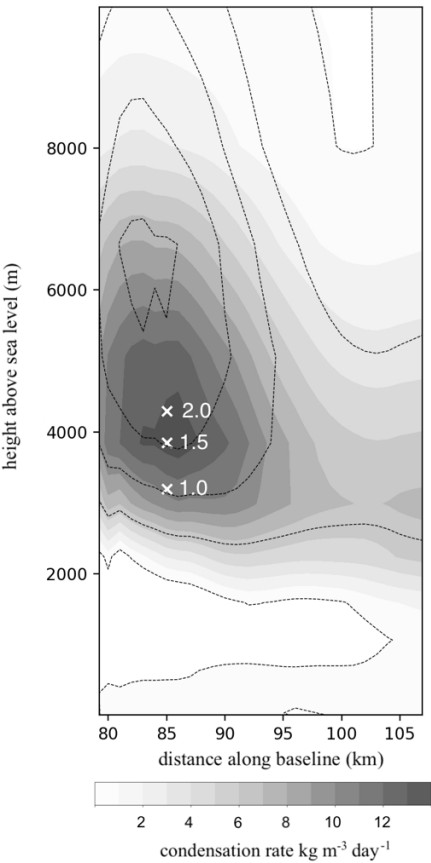

**Figure 7: Local rate of condensation per cubic metre of saturated air per 24 hours with contours of 24-hour mean vertical velocity from UKV. Marked are the start point of precipitation particles arriving closest to Honister with average fall speeds of 1.0, 1.5 and 2.0 m s$^{-1}$ during their ice stage, 6 m s$^{-1}$ during water stage.**

A range of precipitation particles with varying sizes, forms and fall speeds are likely to have been present but nominal particles

were traced on the assumption of steady, average fall speeds. Given that the greatest sensitivity in trajectory was to specified



ice fall speed, values of 1.0, 1.5 and 2.0 m s$^{-1}$ were tried, back trajectories from the nearest rain to Honister meeting the column of ascent at 3200 m (– 2.5 C), 3850 m (– 5.5 C) and 4300 m (– 8 C) respectively. The latter two are close to the peak potential condensation / deposition rates as shown in Fig. 7. For the main trajectories shown in Figs. 9 and 11, a 1.5 m s$^{-1}$ fall speed was specified for ice particles. This is predicated on an average between 1 m s$^{-1}$ for smaller ice precipitation and aggregation snow

flakes, and 2 m s$^{-1}$ for graupel, as found in Doppler radar studies (e.g. Liu and Zheng, 2019 and Makino et al., 2019).

Evaporation of ice particles is particularly efficient (Clough and Franks, 1991) so it is important for their survival that they remain in an environment saturated with respect to ice once formed. It can be seen in Fig. 5 (top) that the whole zone between the Isle of Man and the Lake District along the track of the ice particles is characterized by mean ascent of around 0.2 m s$^{-1}$

which would likely not only ensure the maintenance of such saturation, but enable continued growth in a mixed-phase cloud. As mentioned earlier, the rearward-sloping influence from the principal wave component forced by the Isle of Man, Lake District and Pennines, contributed to this ascent.

6 m s$^{-1}$ was specified for rain fall speed. This is based on figures from Best (1950) who found a mean drop size of 2 mm for

rain whose rate is 10-15 mm h$^{-1}$, a size associated with a terminal velocity of around 6-7 m s$^{-1}$ according to Gunn and Kinzer (1949). No allowance was made in computing trajectories for aerodynamic forces on raindrops, though this is expected to make very little difference to the final trajectory since the rain stage lasts about 4 minutes compared to the overall time of 36 minutes from source to landfall around Honister. Yuter et al. (2006), using a disdrometer, found 0.5 C marked a sharp transition from snow to rain, and this temperature is used to define the melting-level in the precipitation trajectories.


A pre-existing stable layer, with a lapse of around 1 C km$^{-1}$ or less, is evident in the UKV fields, several hundred metres in depth, and lying below the melting layer over the Irish Sea and Isle of Man (Fig. 8). Trajectories show this to move with the average flow, intersecting the melting-level over the Lake District and Pennines, where it weaves around it. Within this zone $\gamma$ is small, amplifying the waves in the melting-level and increasing scope for lee enhancement via the melting-level slope

effect. The vertical resolution of the data from the UKV available for this study is low, with around a 600 m spacing between levels in the vicinity of the 0 C isotherm, though a minimum lapse of 0.6 C km$^{-1}$ is resolved. Consistent with the GWM, we choose an adiabatic lapse rate half way between dry and saturated values at this level, giving $\gamma = 0.075$. It seems that melting of snow from upstream precipitation could well have played a part in forming this near-isothermal layer. Low stability within the boundary layer is very likely due to vigorous mixing by mechanical turbulence forced by strong winds. The resulting

cooling around 1000 m could be a contributory factor in enhancing stability between 1000 m and 2000 m. Regardless of origin, this layer seems likely to have had an important effect on the heavy rain at Honister.





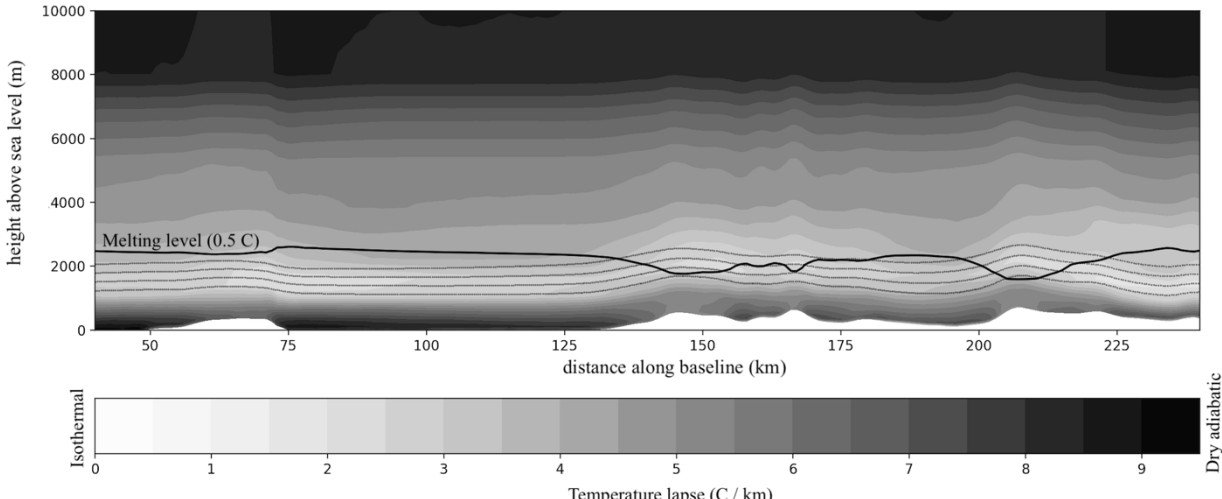

**Figure 8: Temperature lapse (∂T/∂z) in degrees C km⁻¹ together with 0.5 C isotherm and four trajectories through the stable layer.**


Precipitation trajectories given in Fig. 9 were calculated from release points with a vertical spacing of 80 m and arranged in a column spanning the zone of peak precipitation-producing potential as shown in Fig. 7. The most intense areas of ascent and descent are on a small enough scale that they show some evanescence, though still with marked effect on the melting-level, here defined by the 0.5 C isotherm, which mirrors the topography. A large rain-free zone is indicated downwind of where

strong ascent over the rising ground causes both a dip in the melting-level and a levelling out of snow trajectories. Areas of descent then cause the melting-level to rise and snow to fall more quickly to meet it, enhancing rates as trajectories converge. These trajectories are, of course, merely representative of particles with a spectrum of fall speeds for both rain and snow from the values used here of 6 m s⁻¹ and 1.5 m s⁻¹ respectively. Enhancement, calculated on the basis that rain rate is inversely proportional to horizontal distance apart of neighbouring trajectories, gives a factor of 4.0 compared with precipitation rates

over the sea, but at a distance of some 8 km downwind of Honister (at 154 km), which itself lies within the largely rain-free zone. It is of note that this maximum is very close to another very wet site, Thirlmere, which concurrently set a new UK record of 405 mm over two consecutive rain-days.



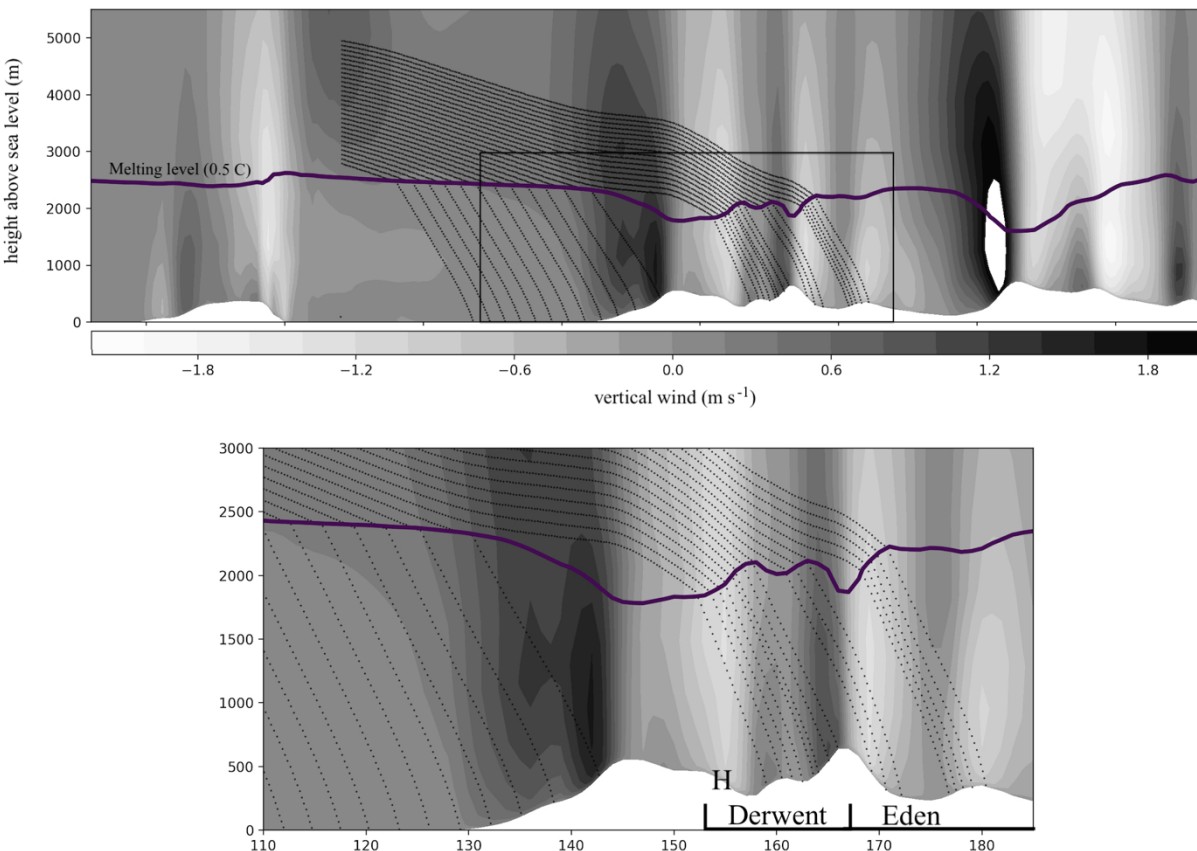

**Fig. 9: Cross section I. 2-D precipitation trajectories originating from column of ascent downwind of Isle of Man between 2900 and 5000 m at 80 m intervals. Advected with UKV time-averaged wind resolved in the plane of the cross section assuming snow fall speed of 1.5 m s⁻¹ and rain fall speed of 6.0 m s⁻¹. Honister (H) and catchment boundaries of rivers Derwent and Eden indicated.**

Although the UKV has a horizontal grid length of 1.5 km for its dynamics, filtering coarsens the resolution of orography (Webster et al., 2003), and unsurprisingly there are discrepancies between the model orography on the line of the cross section and its real-world counterpart, as given in Fig. 10. This shows Honister to lie on a slope which drops from an area of high ground about 500 m above sea level, marked A in Fig. 10. A few kilometres upstream of this is a range of mountains (B), including Pillar at 892 m which is less than 1 km to the north of the cross section. Between these two areas of high ground lies the steep-sided valley of the river Liza (C), with an abrupt drop of some 500 m to valley bottom within around 1000 m horizontal distance. The width of the valley at this point is only about 3 km so can only be expected to be poorly resolved at best, and indeed is scarcely, if at all, apparent in the cross section. Fig. 10 also shows reported rainfall totals. These are given for a 48-hour period since some gauges only report every 24 hours, the event spanning two standard rain days. However it may be inferred that a very large proportion of the values (90% in the case of Honister) given fell in 24 hours.



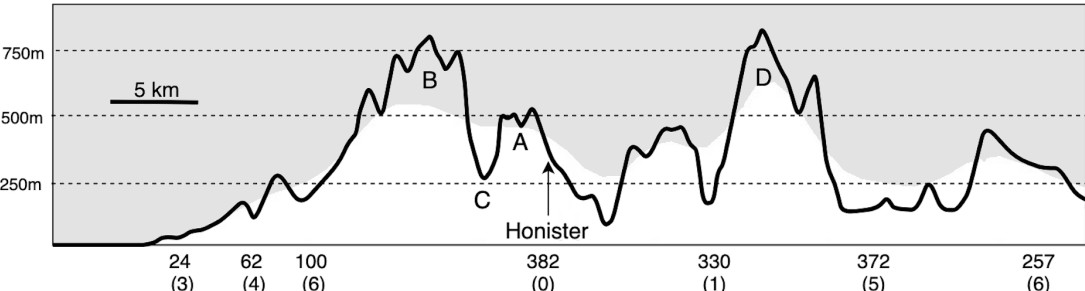

Figure 10: Actual orography of Lake District along line of cross section with model equivalent in white. Letters refer to features mentioned in the text. Numbers at base are 48-hour raingauge measurements in mm from 09:00 UTC 4 December 2015 projected onto cross section with distance normal to the baseline in km in brackets.

The valley of the river Liza (C) widens northwestwards, and to find a UKV cross section which captures it, another (cross section II) was extracted in the same orientation but 3 km to the northwest of cross section I. Precipitation particle trajectories were then calculated as before and are shown in Fig. 11. Here the drop into the valley is apparent (though still rather muted) at around 146 km, giving a marked upward step in the melting-level directly above. The resulting rain falls on the projected position of Honister (150 km) with a similar, though slightly lower, level of enhancement (a factor of 3.8 at 152 km). Thirlmere, at 158 km has a factor of 3.1.

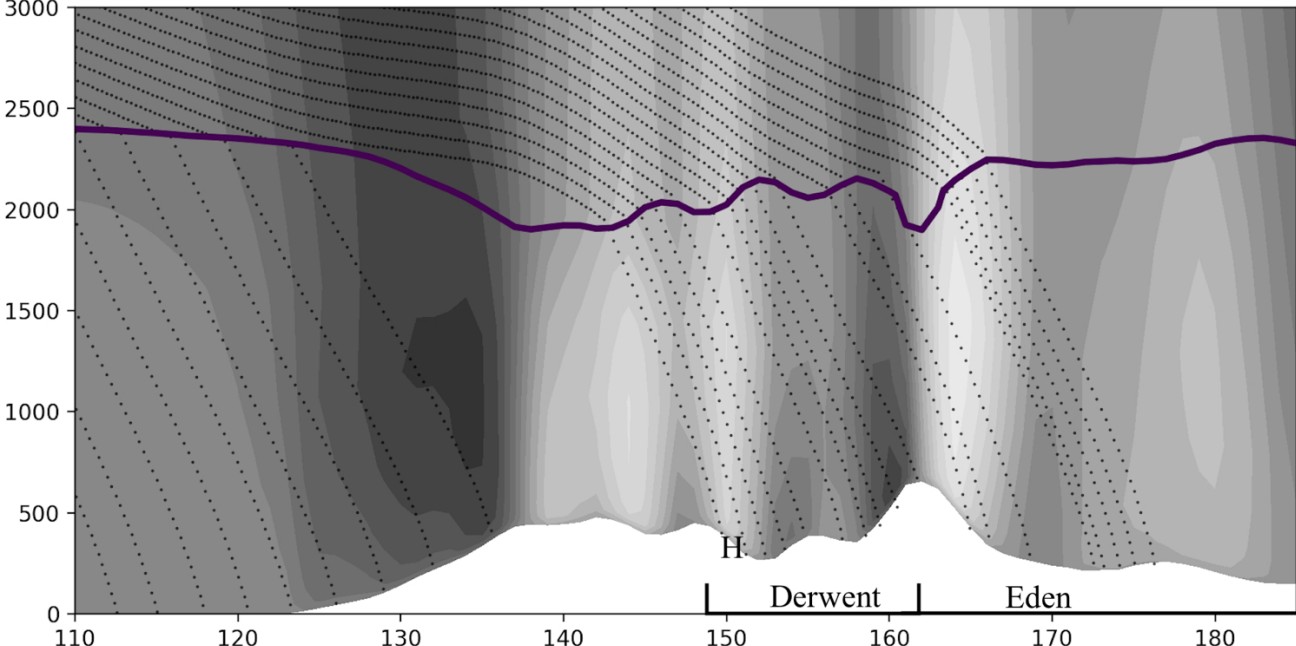

Figure 11: Cross section II. As Fig. 9 but with line of cross section displaced three km northwest.




Input values and results of application of enhancement Eqs. (2) and (4) to cross sections I and II are summarized in the table. The minimum estimated value of $\gamma$ in the stable layer was used since the relatively poor vertical resolution mitigates against sampling the smallest temperature lapses. The values ($E$ = 3.2 and 2.9) fall somewhat short of those suggested by the trajectories ($E$ = 4.0 and 3.8), and in fact the simplified enhancement factor of $w_r / w_s$ worked better. There are a few possible explanations for this. One is that $\gamma$ could be lower than measured due to poor vertical resolution of the available data; another is that the trajectory offset measurements give finite-difference estimates which are subject to errors associated with numerical approximation. Finally, the theoretical treatment which leads to Eqs. (11) and (12) assumes waves and horizontal wind are invariant with height. As seen in Fig. 6, the single wave component of highest amplitude is tilts backwards due to its long wavelength, which could have introduced an extra enhancement factor as found by Stout et al. (1993).

|  | I | II |
|---|---|---|
| $w_m$ | -1.2 | -0.7 |
| $w_g$ | 0.0 | -0.2 |
| $w_r$ | -6.0 | -6.0 |
| $w_s$ | -1.5 | -1.5 |
| $\gamma$ | 0.075 | 0.075 |
| $E_g$ | 3.2 | 2.9 |
| $w_r / w_s$ | 4.0 | 4.0 |
| $E$ | 4.0 | 3.8 |

Table i. Input values and enhancement $E_g$ according to Eq. (2) compared with $E$ values measured from trajectories for cross sections I and II.

Of course, augmentation of rainfall by wash-out of droplets from the feeder cloud, essentially a cloud physics problem, is not included in this, neither is the growth by riming whilst ice particles are in near suspension over the windward slopes as supercooled cloud droplets rise around them. It is of note that the trajectories allow very little rain to fall above 200 m on the windward slope due both to the lifting of snow and the depression of the melting-level in strong ascent, where standard seeder-feeder enhancement might normally be expected to be maximized. This supports the notion that the feeder cloud would be less depleted by wash-out on the windward side than in a situation of a higher melting-level, allowing it to persist onto the lee side of the first ridge. Not only would this offset any tendency for processes to become dry adiabatic on descent, it would allow more wash-out enhancement here.



Both cross sections show marked enhancement towards the eastern end of the Lake District between 170 and 180 km, due to descent from the ridge (D in Fig. 10) which includes such high peaks as Helvellyn (950 m) and Fairfield (873 m). Whilst it might be expected that there should be less scope for low level enhancement of rain through washout given high upstream rainfall, Brotherswater, which lies 5 km south of the line of cross section I at 174 km, recorded 293 mm in 24 hours and 372 mm in 48 hours.

## 6 Conclusion and summary


This study has drawn attention to the importance of gravity wave-induced variations in the height of the melting-level for the modulation of rainfall, which operates via Eq. (2) and, in the limit of an isothermal vertical profile can be said to enhance rain by a factor of rain fall speed divided by snow fall speed (Eqn. (4)). Resulting enhancement differs from conventional mechanisms in concentrating rain preferentially in areas of descent. It has been shown to be increased by:


- A small temperature lapse around melting-level, ideally isothermal
- A large ratio of rain fall speed to snow fall speed
- Strong descent at (therefore strong upward slope of) melting-level
- Strong downslope wind at surface
- Proximity of the melting-level to the ground

Strong descent at melting-level and ground are in turn encouraged by strong horizontal wind speed.

In addition to a melting-level slope effect, Eq. (2) also embodies the modulating effect of the bunching together of precipitation
trajectories through differential advection in wave motion, its relative importance lessening as static stability increases. The melting-level slope effect and bunching effects are of equal size when the static stability is half the adiabatic lapse rate relevant to the setting.

The record-breaking rainfall of December 2015 in the Lake District has been examined. A simple gravity wave model gave a
similar vertical velocity pattern to that of the operational, high-resolution model time averaged over 24 hours, suggesting that rainfall was principally driven by gravity wave motion. A zone of strong, elevated ascent caused by the Isle of Man seems correctly positioned to have been a steady source of seeding ice particles to generate rain over the Lake District. A distinct stable layer is in evidence, intercepting and amplifying undulations in the melting layer which mirror the orography and have a strong focusing effect on rainfall over central and eastern parts of the Lake District, including Honister. Eq. (2) has been
tested and found to give about 80% of the value suggested by rain trajectories, and close to that predicted by the more





approximate Eq. (4). It is acknowledged that the inferences regarding rainfall provenance could be tested more rigorously by a detailed study of the full range of operational model output.

Whilst no precipitation growth calculations have been made, it seems that in situations which favour strong lee enhancement, precipitation on the windward slopes is lessened as snow is prevented from reaching the melting-level by strong ascent. This is likely to reduce wash-out of the feeder cloud upwind of the peak, increasing the potential for such seeder-feeder enhancement downwind of the peak.

Finally, the importance of a model's resolution of narrow valleys has been drawn attention to, since these can be key features for generating intense shafts of rain from the melting level which hit the ground downstream.

**Competing interests**

The author declares that he has no conflict of interest.

**Code availability**

Gravity wave model code (Python 3) is available at https://data.mendeley.com/datasets/4zv2rh892w/1

**Acknowledgments**

The author thanks the Met Office for the data from the operational UKV model and for use of the analysis charts in Fig. 3. This research did not receive any specific grant from funding agencies in the public, commercial, or not-for-profit sectors.

**Appendix A. Derivation of precipitation enhancement equations.**


To examine the scenarios in Fig. 2 and quantify enhancement, we first rewrite Eq. 1 to give the enhancement factor for precipitation which starts as snow from initial height $z_0$:

$$E = \frac{-\partial z_0}{\partial x} \frac{U}{w_s}$$

$$(A1)$$





We now derive an expression for the slope of an isotherm, starting by considering a temperature perturbation at a level due to vertical displacement by height $\Delta z$ of air from another level, which can be given by

$$\Delta T = -\Delta z \left( \frac{\partial T}{\partial z} + \Gamma \right),$$


where $\Gamma$ is the adiabatic lapse rate appropriate to the circumstance, saturated or dry. The resulting change in height of an isotherm, which in stably stratified air is of the opposite sign to the vertical displacement, can therefore be given by

$$\Delta z_i = \frac{\Delta T}{-\partial T / \partial z} = \Delta z \left( 1 + \frac{\Gamma}{\partial T / \partial z} \right)$$


Defining a stability parameter

$$\gamma = \frac{-\partial T / \partial z}{\Gamma},$$


i.e. the environmental lapse rate expressed as a proportion of the adiabatic lapse rate,

$$\Delta z_i = \Delta z (1 - 1/\gamma)$$

$$(A2)$$


Assuming $\gamma$ to be constant in the $x$ direction, the slope of isotherm can be given by

$$\frac{\partial z_i}{\partial x} = \frac{\mathrm{d} z}{\mathrm{d} x} (1 - 1/\gamma) = \frac{w}{U} (1 - 1/\gamma),$$

$$(A3)$$


where $w$ is the vertical wind and $U$ is horizontal wind.

(In certain circumstances, such as transition from a saturated to an unsaturated environment, variation of $\gamma$ along the flow could be a large component of the slope of the isotherm.)






We now examine scenario (b) in Fig. 2, that of simple, steady-state, wave-like trajectories, with horizontal wavenumber $k$ ($= 2\pi$/wavelength) and amplitude (half vertical distance between peaks and troughs) of $A$. The approach taken is to apply Eq. (1) in two stages – firstly to calculate enhancement at the melting-level, then extending it down to the ground, using the identity

$$\frac{\partial z_0}{\partial x_g} = \frac{\partial z_0}{\partial x_m} \bigg/ \frac{\partial x_g}{\partial x_m}$$

560                               (A4)

where $z_0$ is the initial height of generation of a precipitation particle, $x_m$ is its $x$ position along the melting-level and $x_g$ its $x$ position along the ground.


A snow particle's height $z$ can be given in terms of its initial height, $z_0$, a sinusoidal component representing the wave motion of the air, $A \sin kx$ and the particle's vertical displacement relative to the surrounding air parcels (using time $t = x / U$)

$z = z_0 + A \sin kx + \dfrac{w_s x}{U}$

                            (A5)

Following Eq. (A2), the height of the melting-level is given by

$z_m = M + A(1 - 1/\gamma) \sin kx$,

                            (A6)

where $M$ is its mean height.

Where snow encounters the melting-level, $z = z_m$ and Eqs. (A5) and (A6) are equated.


$$z_0 + A \sin kx + \frac{w_s x}{U} = M + A(1 - 1/\gamma) \sin kx$$

$$\Rightarrow z_0 = M - \frac{w_s x}{U} - \frac{A}{\gamma} \sin kx$$

The variation of height of origin $z_0$ with $x$ position along the melting-level, $x_m$, is given by


$$\frac{\partial z_0}{\partial x_m} = \frac{-w_s}{U} - \frac{kA}{\gamma} \cos kx$$





We follow linearised gravity wave theory in ignoring the product of perturbation components so that the vertical wind due to wave motion at any point is given by the product of the mean horizontal wind $U$ and the slope of the waveform at that point, i.e.


$$w = U \frac{\partial A \sin kx}{\partial x} = kAU \cos kx,$$

so we can restate the relationship as


$$\frac{\partial z_0}{\partial x_m} = \frac{-w_s - w_m/\gamma}{U},$$

$$(A7)$$

where $w_m$ is the vertical wind encountered by a snow particle at the melting-level.


Following Eq. (A1), we multiply Eq. (A7) by $- U/ w_s$ to give enhancement over the base rate at the melting-level.

$$E_m = 1 + \frac{w_m}{\gamma w_s}$$

$$(A8)$$


Since $w_s$ is negative, it can be seen that overall rate is increased where vertical wind is downwards at the melting-level, and more so when snow falls more slowly and where $\gamma$ is smaller, i.e. temperature lapse is closer to isothermal. If $E_m$ is negative, it is interpreted as snow diverging from that level because the combination of the melting-level sinking along the flow and upward vertical velocity is too great for the snow to overcome, as seen immediately downstream of melting-level peaks in Fig.

2(b). Provided $\gamma > 0$, i.e. temperature decreases with height, the condition for this is

$$w_m > -\gamma w_s$$

To calculate the enhancement as measured in a gauge we must follow the raindrop from the melting-level down to the ground.

On the way rain will be subject to further intensity change due to the bunching effect, and the modulation along a non-horizontal surface must be corrected to allow for the horizontal measuring plane of a rain gauge. We now set out to take account of these factors.





In the scenario of stationary gravity waves forced by topography, the height difference between ground at $x = x_g$ and melting-level at $x = x_m$ can be given by

$$h = M + A(1 - 1/\gamma)\sin kx_m - A \sin kx_g\,,$$

(A9)

where $M$ is the mean height of the melting-level, the second term represents its variation around the mean height and the last the variation of orography forcing the wave.

From the raindrop's perspective, the vertical distance covered whilst falling to the ground can also be given by

$$h = \frac{-w_r}{U}\left(x_g - x_m\right) + A \sin kx_m - A \sin kx_g$$

(A10)

the first term being the vertical distance covered by the rain falling relative to the surrounding air, the second and third terms accounting for the difference in phase of the wave motion between the melting-level (second term) and ground (third term). By equating Eqs. (A9) and (A10) we can find an expression for $x_g$, the location where the rain emanating from $x_m$ on the melting-level hits the ground.

$$\frac{w_r}{U}\left(x_g - x_m\right) = -M + \frac{A}{\gamma}\sin kx_m$$

$$\Rightarrow x_g = x_m + \frac{U}{w_r}\left(\frac{A}{\gamma}\sin kx_m - M\right)$$

The variation of $x_g$ with $x_m$ can therefore be given by

$$\frac{\partial x_g}{\partial x_m} = 1 + \frac{kAU}{\gamma w_r}\cos kx_m$$

Using $w_m = kAU \cos kx_m$, leads to

$$\frac{\partial x_g}{\partial x_m} = 1 + \frac{w_m}{\gamma w_r} = \frac{\gamma w_r + w_m}{\gamma w_r}$$



Using this result with Eqs. (A7) and (A4) gives


$$\frac{\partial z_0}{\partial x_g} = \frac{-w_r}{U} \frac{(w_m + \gamma w_s)}{(w_m + \gamma w_r)}$$

Following Eq. (A1), enhancement along the ground relative to unmodulated precipitation rate is given by multiplying by $- U$ / $w_s$ to give


$$E = \frac{w_r}{w_s} \frac{(w_m + \gamma w_s)}{(w_m + \gamma w_r)}$$

We are nearly there now, but since $\Delta x$ is the $x$ increment along the sloping ground rather than horizontally, the expression has to be corrected by a factor which converts it to the amount which would be measured in a standard, horizontally aligned rain

gauge.

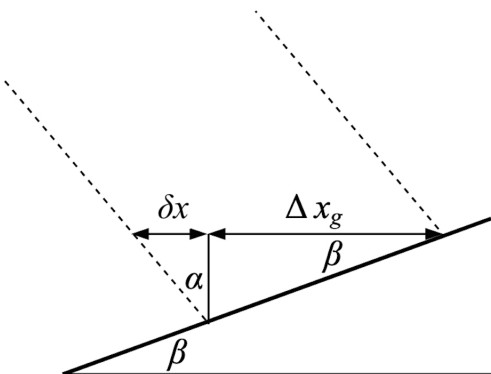

**Figure A1: $\beta$ is the local angle of orography to the horizontal, $\alpha$ the angle of fall of rain with the vertical, $\Delta x_g$ the horizontal distance between the locations that adjacent streams of raindrops reach the ground.**


To correct $E_g$, it must be multiplied by

$$\frac{\Delta x_g}{\Delta x_g + \delta x}$$

We solve for $\delta x$ by using

$\delta x = \Delta x_g \tan \alpha \tan \beta$                                                                    ;



$$\tan \beta = \frac{w_g}{U}$$

$$\tan \alpha = \frac{U}{w_r + w_g},$$

where $w_g$ is vertical velocity at the ground, and find that the enhancement factor $E$ must be multiplied by

$$\frac{(w_r + w_g)}{w_r}$$

to give a total enhancement factor for a horizontal raingauge on the ground

$$E_g = \left(\frac{w_r + w_g}{w_s}\right) \left(\frac{w_m + \gamma w_s}{w_m + \gamma w_r}\right)$$

(A11)

as represented in Fig. 2(b).

In order to quantify the separate effects of melting-level slope and bunching as represented in Figs. 2(c) and (d), we can go through the same process but with some modifications to the input equations.

Bunching-only enhancement at the ground (as in Fig. 2(d)) is derived by eliminating terms in variation of melting-level in Eqs. (A6) and (A9) to give

$$z_m = M$$

and

$$h = M + A \sin k x_m - A \sin k x_g$$

This removes dependence on $\gamma$ and leads to


$$E_b = \left(\frac{w_r + w_g}{w_s}\right) \left(\frac{w_m + w_s}{w_m + w_r}\right)$$

(A12)

Melting-level slope enhancement at the ground (Fig, 2(c)) can be derived by removing terms representing sinusoidal movement
of air in Eqs. (A5) and (A10) to give





$$z = z_0 + \frac{w_s x}{U}$$

and


$$h = \frac{-w_r}{U}(x_g - x_m) - A \sin k x_g$$

which leads to

$$E_{mls} = \left(\frac{w_r + w_g}{w_s}\right)\left(\frac{(1-\gamma)w_m + \gamma w_s}{(1-\gamma)w_m + \gamma w_r}\right)$$

710                                                                 (A13)

It can be seen from Eqs. (A12) and (A13) that the bunching and melting-level slope effects are equal when $\gamma = 0.5$, i.e. the environmental lapse rate is half the adiabatic lapse rate. From Eq. (A2) it is evident that, in this instance, the melting-level

exactly mirrors the vertical displacement of air parcels, giving the configuration shown in Fig. 2.

The melting-level slope effect dominates as $\gamma$ approaches zero. When $\gamma = 0$, it gives the full enhancement

$$E_{mls} = E_g = \left(\frac{w_r + w_g}{w_s}\right)$$

(A14)

Finally, these results can be generalized by aggregating a Fourier series of sine functions to different velocity patterns, periodic or otherwise. The treatment can also be directly generalized to any vertical velocity pattern which is a continuous function of $x$, $f(x)$. For example, the trajectory of the snow particle (Eq. (A5)) then becomes


$$z = z_0 + f(x) + \frac{w_s x}{U}$$

and the position of the melting-level for the fully adiabatic case (Eq. (A6))

$$z_m = M + (1 - 1/\gamma)f(x)$$





Using $w = U f'(x)$, where $f'(x)$ is the derivative of $f(x)$, leads to Eq. (A7), showing that the enhancement equation works equally for a straight, sloping melting-level, or one with a complex, polynomial shape.

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
