# Peer review of "Rainfall enhancement downwind of hills due to standing waves on the melting-level and the extreme rainfall of December 2015 in the Lake District of northwest England"

_EGUsphere, 2023_

## Author Comment (AC1)

*Anonymous referee #1 comments are in black and author responses are in purple italics.*

General comments:

The author introduces an interesting thought experiment regarding the role of orographically-induced gravity waves on precipitation distribution across the Isle of Man and the Lake District of northwest England.  Unfortunately, the data presented in support of these ideas is insufficient to justify publication in a peer-reviewed journal.  The author relies on relatively coarse operational model output that does not include any moisture variables.  Additionally, verification of this model output is virtually absent, which significantly limits its credibility, especially when discussing orographic precipitation processes.  A separate, higher resolution gravity wave model is also employed, whose results seem to be consistent with the operational model output.  However, this does not mitigate the aforementioned lack of model validation.  It also does not mitigate the absence of moisture variables in the analysis dataset.  The author has to make numerous assumptions in his analysis to get around this limitation.  I find many, if not most of these assumptions to be dubious.

Ultimately, the author makes assertions that are not supported with credible evidence.  As a result, my recommendation is to reject the manuscript for publication.

*Thanks very much for reading my paper and taking time to comment on it. I address these general comments beneath in response to your specific points below. As a general point, I wanted specifically to put cloud microphysical processes to one side in my treatment, not to minimise their importance, but to indicate that I am elucidating a process which arises due to the dynamics/kinematics; that what I describe occurs independently of much of the microphysics, can be considered quite separately as a complementary process, and does not require high vertical resolution model data to examine. Incidentally, the main point of the paper was to draw attention to the rainfall intensification/redistribution mechanism, illustrating it graphically and deriving an analytical quantification for it, since I don't think this has been done before. You don't explicitly comment on the main analytical result, that rain rates can be increased by a factor approaching rain fall speed divided by snow fall speed through the effects I elaborate.*

1.  sections 2 and 3:  The author goes into great detail about how gravity waves and an oscillating melting layer can impact the spatial distribution of precipitation across orography (i.e., Figs. 1-2; Equs. 1-4).  He calls the areas of precipitation where the "trajectory" lines are closer together more "intense" than the areas where the trajectory lines are farther apart.  If precipitation intensity is based on an areal integration, this may be an appropriate interpretation.  However, precipitation intensity is typically based on precipitation rate, which is a mass flux for a given vertical column.  The only way that precipitation intensity can be enhanced is by adding mass to the volume of hydrometeors through cloud microphysical processes.  The author is not really addressing precipitation enhancement; rather, he is addressing precipitation redistribution.

*But if you redistribute a given mass of precipitation into a smaller area, its intensity necessarily increases whether the area is the size of a raingauge or a river catchment. There seems to be a substantive difference of opinion between us here since I don't agree with your assertion that the only way precipitation intensity can be enhanced is by adding mass through microphysical processes.  I think I am using the same definition of 'intensity' as you, i.e.*

*precipitation rate as commonly represented in mm/hr. Where precipitation trajectory lines become closer together, precipitation which originated over a given depth in a column is concentrated into a column of narrower cross-sectional area, with increased water mass flux relative to that which would occur in the absence of the effect. Mass flux is measured in kg per metre squared per second (numerically equal to 3600 times the mm/hr value), so is defined in terms of a mass passing through an area (rather than a volume) per unit time. If you reduce the area over which it falls, a given mass of precipitation results in a greater intensity. Perhaps I did not expand on this point fully enough (and I hope I'm not labouring it here), but I had assumed that it was demonstrated in Fig. 1 and accompanying text lines 56-59. Unlike enhancement through cloud washout, the total area integrated water mass is unaltered; rainrate and therefore rainfall totals are just intensified by being concentrated in a smaller area.*

*I wonder whether the use of the word 'enhancement' is the cause for confusion, since in the context of rainfall it is often used in specific sense of adding mass to existing hydrometeors e.g. through washout. But here I was simply using the word consistently with its definition as 'intensification' or 'increase' in the same way one might talk about enhancement of convection or enhancement of a jet. Perhaps I should have made all this clear or used another expression such as 'rainfall intensification' or 'rainfall focussing' but I did try in several places to draw the distinction with microphysical mechanisms. Whatever it is called, I think I have presented strong evidence that it occurs and can have a dramatic effect on measured rainfall rate, increasing it severalfold.*

2. sections 2 and 3: Does the author have any observational evidence to support the notion that a melting layer can oscillate as he hypothesizes? For example, are there any radar studies that show a bright band that oscillates in such a manner?

*In case there is any confusion I did not wish to imply that the melting level oscillates in the examples given, but that it varies in height with horizontal distance – the examples I give are steady-state. I do not have specific observational evidence but it is an inevitable consequence of adiabatic temperature response to vertical velocity in a stable airmass where temperature decreases with height, and can be verified with a simple thought experiment. It also comes out explicitly in my analytical treatment (e.g. eqn A6) and is clearly evident in the model output shown, where the close mirroring of the orography by the melting level would be too coincidental to be ascribed to anything else.*

3. L274-279: This paragraph outlines the very limited nature of the data available to the author. This data was in the form of 24-hour mean fields from a single operational model simulation. The data was limited to horizontal and vertical winds and temperature. No moisture variables were available (i.e., water vapor, precipitating ice and/or liquid water). While the horizontal resolution was 1.5 km, there were only 14 vertical levels (~600 m resolution near the melting level). No attempt at validating the model output is evident. This dataset is clearly insufficient for addressing the processes discussed by the author.

*I address the two points made here, (1) absence of verification and (2) sufficiency of model output for addressing the processes discussed.*
*(1) I concede that no attempt was made at validating the model output, but this would be difficult given the absence of observations of vertical velocity. Given that it is a short period forecast it is considered that the model is unlikely to have been incorrect in the overall signal, I have taken it as 'truth' for the purposes of revealing likely flow and thermal*

*structure. I agree that it would have been good to have the model rainfall predictions for verification but unfortunately I cannot get them. I should have stated more clearly that I was taking the model as a proxy for 'truth' and justified that choice. As such it is an internally consistent source of data which supports my mechanism in generating significant intensification of rain in the right area and qualitatively explains the west-east variation in reported rainfall values as given in Fig. 10. It seems more than coincidental that there was extreme rainfall at Honister. The advantage of such a simple model with no cloud physics (beyond specified fall speeds) over a sophisticated cloud physics model is that the output results largely from effects treated in the analytical approach, which is what is being tested.*

*(2) I should clarify that the UKV model from which output was drawn had 70 vertical levels and was a state-of-the-art mesoscale model which was used to inform the successful issue of flood warnings at the time. However, I was unavoidably using a dataset with only 14 of those levels. I do not agree that this dataset is insufficient for addressing the processes I discussed because the output reveals well-defined structures in the vertical velocity field which were much larger than the vertical resolution available, so it seems to me that it is not a serious shortcoming; whilst being potentially problematic for determining lapse rate over small distances, as I drew attention to, any such insufficiency would have the effect of lessening the mechanism I was investigating. Despite this, there is clearly a lot of detail evident in the model melting level.*

*So I contend that the model output was adequate for my purposes, that of tracing nominal precipitation trajectories, thereby giving reasonable corroboration for my derived equations, and revealing large scale dynamical structure to compare with the gravity wave model. If I had been doing a study of microphysical precipitation mechanisms then this would clearly be different and the absence of humidity data would obviously have been a serious shortcoming.*

4. L333-338: This paragraph makes a sweeping assertion: "However, the generally good correspondence between this output and the UKV, despite all the caveats, along with the remarkable steadiness of rain rate at Honister, strongly suggests that gravity waves were the main driver for vertical velocity over the period of extreme rainfall and therefore for the rain itself." The operational model output and corresponding gravity wave model output do not provide sufficient evidence to support this assertion. In particular, the lack of moisture variables eliminates evidence of possible alternative explanations based on cloud microphysics. The author does not present evidence about the depth of precipitation and whether there are hydrometeors aloft that could be influenced by the vertical velocity patterns described. For all we know, the precipitation could be very shallow in nature.

*Clearly cloud microphysical processes are indispensable to precipitation production, but my starting point is that significant ascent (upward motion) is necessary to produce the rapid cooling and supersaturation required for significant precipitation, and that once formed the hydrometeors would move largely with the wind – both these points seem uncontroversial to me though perhaps I should have expanded on them. I tried to indicate (e.g. line 340) that I was not dealing with the microphysical part of the problem, but acknowledge its importance as an additional factor (e.g. line 145, 461). Large scale dynamic / kinematic explanations for drivers of precipitation are in general clearly separated from the microphysical; your point about alternative, microphysical explanations to the gravity wave explanation seems to be at cross-purposes, in the way that someone might say of another situation 'how do you know that it was the cold front which caused the rain and not microphysical processes?'. I would*

*go further and say that for this case cloud microphysics cannot provide a good alternative explanation for the cause of the rain because without strong ascent to provide rapid generation of solid/liquid water, no microphysical process could generate such large totals.*

*By the same token it seems highly unlikely that the precipitation was very shallow in nature since all empirical evidence I'm aware of indicates that deep layers of precipitation-producing cloud are necessary for very heavy rain. Perhaps I should have made all this clearer.*

5. L461-463: The author states: "Of course, augmentation of rainfall by wash-out of droplets from the feeder cloud, essentially a cloud physics problem, is not included in this, neither is the growth by riming whilst ice particles are in near suspension over the windward slopes as supercooled cloud droplets rise around them." This apparent "disclaimer" does not make any attempt to diminish the significance of these processes. It is quite possible that these processes are the dominant factors in the precipitation distribution associated with the case.

*Yes, I certainly agree that cloud microphysical processes are very likely significant in the enhancement of precipitation in this case and could be the dominant factors for intensity if not distribution. I'm not sure what you mean by 'disclaimer' but it was not meant in any way to deny the contribution of cloud physical processes, rather to indicate again the limits of my quantitative treatment. They have generally been well documented and their operation is quite well understood. I merely wanted to point out that before one even considers them, the mechanism to which I have drawn attention, which to my knowledge has not been described before, is likely to have intensified precipitation by a significant factor. Whatever enhancement multiplication factor might be attributable to microphysics, it is acting on top of the effect which I have elucidated.*

6. L494-496: The author reasserts an unsupported conclusion that the rainfall in the case study was "principally driven by gravity wave motions".

*If one accepts that heavy precipitation must be driven by ascent, and that a pure gravity wave model reproduces the primitive equation model's pattern of ascent quite well, then I hope I have presented quite strong, if not overwhelming, evidence that the precipitation was caused by gravity waves. Of course one could also tackle the problem from a different angle, take the dynamical processes as a given and consider the precipitation as being caused by a whole train of microphysical processes from ice nucleation, sublimation, riming, aggregation, cloud washout etc. for which you would need a much more complex numerical model, which is perhaps what you are thinking of. This was not my purpose. The effect I'm investigating is complementary and, as far as I'm aware, has not been described before. I'm not setting it up as a rival explanation for rainfall enhancement, just an additional factor which reveals a surprising quantitative relationship and, I hope, an illuminating conceptual model.*

---

## Author Comment (AC2)

*In order to address issues relating to the use of output from short period forecast of the UKV as 'truth', and to the absence of moist variables, the author has made use of the Copernicus European Regional ReAnalysis (CERRA) dataset, produced by a ~ 5.5 km formulation of the Harmonie Aladin assimilation-NWP system with boundary conditions from ERA 5 (Schimanke et al. 2021). Essentially the same timeframe was used, with 24 hour means calculated up to 18Z 15th December, but 19Z in the case of moist variables due to a constraint on write-out times. The CERRA reanalysis output may be deemed more acceptable as a source of pseudo observations, though its lower resolution makes it unsuitable for the small-scale detail important to the case. The output provides support for the UKV forecast used in the paper, and for inferences drawn about snow production in the column downstream of the Isle of Man.*

**1. Verification**

*See below a temperature-height plot from the UKV and CERRA systems for a point in the Irish Sea. This shows a close correspondence between the fields from the two models, giving confidence in the UKV output. Choosing a point over land gives more noticeable differences principally in the boundary layer due to the coarser orography in CERRA. Note that CERRA also supports the near-isothermal layer between 1500 m and 2000 m drawn attention to in the paper.*

[Figure]

*Fig. 1 CERRA (crosses) and UKV (circles) temperature vs height for point in Irish Se. Mean during 24 hrs to 18Z 5 Dec 2015.*

**2. Snow**

[Figure]

*Fig. 2 Left: CERRA Vertical profile of snow content (24 hr mean to 19Z 5 Dec 2015) at distance 85km on cross section baseline. Right (From Fig. 7 in paper). Vertical velocity (contours) and estimated condensation rate (shaded) from UKV 24 hr mean to 18Z 5 Dec 2015. Origin points of precipitation falling as rain near Honister with different snow fall speeds indicated.*

*CERRA snow content is defined as 'the mass of snow (aggregated ice crystals) produced from large-scale clouds that can fall to the surface as precipitation'. Its profile (located just downwind of the Isle of Man) agrees qualitatively well with the pattern of condensation rate estimated from w dqsat/dz in the paper, peaking at 4000 m, decreasing slowly above, more quickly below. Whilst values seem numerically quite small, it should be remembered that these are interpreted as ice seeds forming in ascent and growing significantly downstream by riming etc. and that some of the condensation is partitioned into cloud ice, cloud water and rain.*

*Schimanke S., Ridal M., Le Moigne P., Berggren L., Undén P., Randriamampianina R., Andrea U., Bazile E., Bertelsen A., Brousseau P., Dahlgren P., Edvinsson L., El Said A., Glinton M., Hopsch S., Isaksson L., Mladek R., Olsson E., Verrelle A., Wang Z.Q., (2021): CERRA sub-daily regional reanalysis data for Europe on single levels from 1984 to present. Copernicus Climate Change Service (C3S) Climate Data Store (CDS), DOI: 10.24381/cds.*

---

## Author Comment (AC3)

Referee's comments in black. Author's response in purple italics

**Referee 2 comments**

**Major Comments**

*Introduction:*

Several key references, and some description therein, is missing in the introduction's description of the state of knowledge about orographic precipitation, its relationship to standing gravity waves, and the undulations of the melting layer. Regarding the first two, I refer the author to two widely cited review papers on orographic precipitation, Roe 2005 and Houze 2012, and two book chapters, Colle et al. 2013 and Stoelinga et al. 2013 (both are chapters from the same book). Regarding the melting layer and its variations in height with respect to the terrain, Minder et al. 2011 thoroughly explores the contribution of three mechanisms, two of which are described in this paper (albeit through a somewhat different lens). Although Minder et al. (2011) focuses on an idealized case where the snow line intersects the terrain, the discussion of mechanisms that modulate the altitude of the melting layer are highly relevant to this paper. Significantly more attention to prior literature and discussions about the mechanisms at play is necessary for this paper to adequately address its contribution.

*The author thanks this referee for their careful reading and the many useful suggestions which have improved the paper.*

*Line numbers refer to those in revised manuscript.*

*I had not intended the introduction to cover the whole subject area of orographic enhancement of precipitation but agree there is value in citing the papers suggested in order to provide a comprehensive background, and have included references to all the papers suggested, mainly in the introduction. I have introduced ideas about the melting level relating to the paper of Minder et al. in the discussion in section 4.1 lines 536 onwards. I have also cited Stoelinga et al., 2013 in this section and expanded the discussion of diabatic effects of melting snow (paragraph line 549 onwards. This has prompted me to rework the analytical treatment in the Appendix to include an explicit diabatic term D as explained below).*

*Section 2 and associated Appendix:*

These sections need considerable rewriting and reorganization to more clearly state why each equation is shown, how it is derived, and critically what assumptions are made in its derivation. In addition, much of the language surrounding the equations is vague and/or conversational; this section should be explicit and extremely plain with its language, for clarity.

*To address the charge of conversational language in section 2 (now section 4) and the Appendix, I have reworded sections in various ways, including eliminating use of the pronoun 'we' as in 'we find that...' by using constructions in the passive voice, e.g. 'it is found that...'. I have replaced the conjunction 'so' where appropriate with the more formal equivalent 'therefore'. I do not list all the lines at which this is done.*

*I have also reworded Section 4 and the Appendix with a view to clarification. The equations are all derived from first principles and there is a balance to be struck between concision and*

*ease of following, but I have added extra explanatory text, e.g. lines 670- 671, 677 – 679 and Fig. A1 where positions and heights referred to in equations are shown diagrammatically. I have expanded the assumptions listed, e.g. lines 708 – 709, and for clarity and completeness extended the previously adiabatic treatment to one which caters for diabatic effects by introducing term D, line 736 onwards. This makes the treatment somewhat more complete and changes subsequent equations including the results in equations (5) and (A11) but leaves the overall isothermal, limiting case unchanged (equation 6).*

*Paper structure*

The paper begins with its derivation of its precipitation trajectory mechanisms, followed by some discussion of those mechanisms, and then goes into a case study. This structure seemed back-to-front to me, and the story would have been significantly more clear had the paper been structured as follows: Following the introduction, the data and methods for the case study analysis should be clearly laid out including a description of the two models (UKV and GWM) and any other data used in the paper as well as a description of the particle trajectory software/process used (perhaps this is part of the GWM but it is not clear). Then, the case study could be described, and used to motivate the derivation of undulating melting layer+GW bunching enhancement. The last results section should then apply the enhancement to the case study (pages ~17-21 of the current paper) with some discussion of the value added of the new method, and the paper can then end with a conclusions section.

*I have followed all these suggestions. The reordering of the paper structure necessitates rewording in multiple places to change backward and forward references to other parts of the paper, too many to list here. The data and methods section introduces a new source not included in the original submission, the Copernicus reanalysis data (CERRA, lines 64-69) which provide evidence for cloud and precipitation data relating to the case and also some verification of the UKV fields. Some discussion has been included, as suggested, in the results section (4.2) focussing on its worth as a conceptual model of a category of rainfall enhancement which seems to have been overlooked (lines 616 – 620)*

*Figure use and reference*

In general, the figures should be referred to at specific sections of the text when they are discussed.

*I have made some changes, including dropping the early reference to Fig. 8 before 5,6,7 (numbers now all change due to change in ordering).*

**Minor Comments**

1. 25: I suggest adding 'mechanisms' to the text 'One of the first to be described...' so that it reads 'One of the first mechanisms to be described...'

   *Done.*

2. 26: I suggest adding 'moist' to '... replenished by the ascent of air...' so that it reads '... replenished by the ascent of moist air...'

   *Done*

3. Figure 1: I suggest you add a vector indicating the wind is blowing from the left.

*Done*

4. 57-58: This sentence requires considerable assumptions, e.g. that the evaporation doesn't change across the interface, etc. More attention should be given to the assumptions made prior to each assertion.

   *Assumptions elaborated, including evaporation, sublimation and melting (Lines 388 – 390).*

5. 70: Is $w_s$ assumed to be negative or does the negative sign before $w_s$ in the equation capture the downward direction (i.e., the sign conventions used are not clear)?

   *Clarified. $w_s$ is negative so $– w_s$ is positive. (Line 409)*

6. Figure 2: I believe this figure is intended as a toy schematic for teasing apart the mechanisms, but this is not clearly described, and as such it is simplified to the point of being incorrect.

   *I have clarified the assumptions made lines 420-425.*

7. 81-83: These three sentences need some revision for clarity.

   *Clarification as above and Lagrangian model expanded upon lines 86-90.*

8. 96-98: This should refer back to Figure 1.

   *Done, lines 441 (though now Fig. 1 is Fig. 13)*

9. 104: Stout et al. 1997 should also be cited here.

   *Done*

10. 129: 'So the magnitude of modulation...' this is extremely conversational and needs to be revised for clarity.

    *Deleted since not important.*

11. 146-149: Where has this analysis been done? Is this testing not shown?

    *More explanation lines 499 – 505, and Fig. 15 added which shows scatter plot of analytical vs modelled results. Not totally satisfactory since I didn't find a way to highlight how variation in individual parameters influences different scatter plot*

*results but it conveys the overall sense of accuracy, along with quoted mean absolute error stats.*

12. 220-224: This section of text poorly described and needs expansion for clarity.

*Deleted since not important.*

13. 251: This is, I believe, the first introduction of the UKV, and it needs to be defined.

*Now introduced in new Data and methods section lines 58-60.*

14. 254: A map of the analyzed rainfall should be included as one of the figures for the case study.

*A map of reported raingauge measurements colour-coded according to amount has now been included (Fig. 2). This is not analysed with isohyets, but I feel this sort of analysis can be misleading because we don't know what is happening in individual peaks and valleys, and colour coding goes some way to visualising spatial distribution at a glance. Time evolution of rain at Honister also included.*

15. 274-279: This text which describes the UKV model should be moved into a section where data and methods are described (adjacent to the GWM description). Why were moist variable data unavailable?

*Done. Moist variable data were not presented to me as a menu option when extracting data. The shortfall has been supplied by using CERRA moist data, as in Figs. 6 and 7.*

16. 293-294: 'Note that only the region...' is not a strong start to a new section. Transitions should be used to make the paper more readable.

*Deleted since not important.*
17. 300: Figures 5, 6, and 7 were not referenced before this reference to Figure 8.

*Early reference to Fig. 8 dropped.*

18. 308: This satellite imagery is not shown.

*Reference to satellite imagery dropped.*

19. 326-331: These experiments are presumably not shown; 'not shown' should be explicitly stated.

*Done (line 216).*

20. 336-338: Since these are 24-hour averages, an alternative hypothesis would be that any diabetic/other effects that generated vertical velocity and precipitation occurred randomly through the domain at shorter time intervals and thus when averaged, their signal was largely removed.

*Done (line 228).*

21. 370-372: This sentence needs revision for clarity.

*Done (line 269-273).*

22. 381: principal->principle

*As far as I can see the original is correct, so this is left unchanged (now line 282).*

23. 413-416: The paper should refer to its equations for the enhancement calculation.

*Done (line 327).*

24. 526-545: This section of the appendix provides an example of what I'm suggesting in my Major Comment regarding Section 2 and the Appendix: This section describes an expression for the slope of an isotherm, but does not motivate this derivation by noting that it will be applied for a specific isotherm, the melting level. It also needs a bit more thorough defining and discussion for each equation shown (and for any inferences made between equations).

*Along with the rewording to make it seem less conversational, I've tried to address this criticism by adding more explanation at various points through the Appendix to explain the approach, e.g.*

*Line 670, rationale added for derivation of expression of isotherm.*

*Lines 709 – 713, description added to prepare the reader for the derivations which follow.*